# Apple and Pear Model for Optimal Production and Fruit Grade in a Changing Environment

**Miguel L. Sousa** [1,*] **, Marta Gonçalves** [1] **, Délia Fialho** [2] **, António Ramos** [3] **, João P. Lopes** [4] **, Cristina M. Oliveira** [4] **and J. Paulo De Melo-Abreu** [4,*]

1 Instituto Nacional de Investigação Agrária e Veterinária, I.P.—Estação Nacional de Fruticultura Vieira Natividade, Estrada de Leiria, 2460-059 Alcobaça, Portugal
2 TriPortugal & Frutus—Estação Fruteira de Montejunto CRL, EN 366, km 4, 2550-452 Peral, Portugal
3 Escola Superior Agrária, Instituto Politécnico de Castelo Branco, Av. Pedro Álvares Cabral 12, 6000-084 Castelo Branco, Portugal
4 LEAF—Linking Landscape, Environment, Agriculture and Food Research Center, Associated Laboratory TERRA, Instituto Superior de Agronomia, Universidade de Lisboa, Tapada da Ajuda, 1349-017 Lisbon, Portugal
* Correspondence: miguel.leao@iniav.pt (M.L.S.); jpabreu@isa.ulisboa.pt (J.P.D.M.-A.)

**Abstract:** Apple and pear crops are very important to the rural economy of Portugal. Despite significant improvements in productivity and quality, due to the introduction of new management techniques, model-based decision support may further increase the revenue of the growers. Available simulation models of orchard growth and production are scarce and are often highly empirical. This study presents a mechanistic model for the simulation of productivity and fruit grade of apple and pear orchards under potential and water-limited conditions. The effects of temperature extremes and rain on fruit set are addressed. The model was validated on apple and pear datasets derived from extensive experiments conducted in Central and Southern Portugal. Model performance is high and depicts the effect of crop load on productivity and fruit-size grade and the distribution of both crops. A simulation example shows the relationship between productivity and average fruit size for a hypothetical six-year-olc apple orchard. The model herewith presented is a tool that can be used to estimate optimal crop load for maximum revenue and productivity, fruit size distribution, water use, and other variables relevant for pome fruit production.

**Keywords:** crop model; productivity; crop load; canopy photosynthesis; *Malus domestica* Borkh; *Pyrus communis* L; bee activity index; CSS.Pome

## 1. Introduction

Apple and pear crops are very important to the rural economy of Portugal. In 2021, Portuguese apple and pear production ranked eighth and fifth, respectively, in EU27 [1]. Productivity and fruit quality have steadily increased, but some issues need attention for further improvements to occur. Quantitative answers to questions about the degree of intensification, the effect of the training system on productivity and quality, and the optimization of fruit grade and other quality parameters are often site-specific, inaccurate, and/or baseless. Suitable mechanistic-mathematical models are able to handle the complexity associated with these systems, to improve our understanding and provide solutions compatible with the problems currently faced by producers [2].

Few mechanistic models of fruit crop production have been constructed due to the lack of data to calibrate the overall models and all required sub-models, which are needed to represent the main underlying processes. The models PEACH [3], L PEACH [4], and CSS.Pear [5,6] are some of the exceptions. All these models are based on well-established approaches: (i) interception of radiation follows the exponential extinction model [7], modified to account for the discontinuity of the canopy [8,9]; (ii) development uses the

thermal time approach [10]; (iii) canopy photosynthesis uses the "photosynthetic efficiency" approach [11]; (iv) maintenance and growth respirations follow the approaches reported by [12–14]; (v) dry-weight (DW) and leaf-area distributions use fixed-partition factors; (vi) potential transpiration relies on reference evapotranspiration and crop coefficients [15–17] and (vii) actual evapotranspiration is computed from potential evapotranspiration using a reduction factor, which is a function of relative water availability in the soil [17,18].

Global climate change is leading to changes in precipitation, air temperature, solar radiation patterns, and increasing climate variability, which will affect the productivity and quality of pome fruit production. The sustainability of production or the adequacy of certain cultivars in some regions may change. Global climate change is likely to reduce productivity due to high temperatures during flowering and fruit growth, and fruit sunburn is likely to increase [19–24]. Therefore, it is useful to have the support of exploratory tools that can provide adequate answers in an economical and timely manner. We assume that only process-oriented models are able to perform well in environments that differ from those in which they were developed. Moreover, only these models are likely to allow for the comparison of cultivars when specific parameters are input.

After decades of experience and collecting basic data on apple and pear crops, we conducted detailed experiments in Central and Southern Portugal for twelve years. This study presents a model named CSS.Pome. It was constructed and validated using the data generated across locations and crop years. The model simulates water-limited production and also takes into account the damage caused by extreme events, such as frost damage, and the effects of cold or heat on the fruit set. Some processes have been described in detail, such as the interception of radiation and carbon assimilation, due to their expected ability to better adapt to changing environmental conditions and present and future crop varieties. Canopy photosynthesis relies on a sub-model that includes the kinetics of leaf photosynthesis [9,25,26].

This study introduces the model, the evaluation for apples and pears, and illustrates its use in the quest to optimize crop fruit load.

## 2. Materials and Methods

### 2.1. Model Description

The model CSS.Pome simulates the growth and production of apple and pear crops. The model is dynamic, mechanistic, and deterministic, and requires a fair number of parameters as input, although there are some more advanced parameters that can be altered, but usually do not have to be. It simulates one or more years of tree development and the growth, productivity, and water consumption of the crop, and covers crop and fruit size distribution in custom-defined size classes. The model is coded in Visual Basic.

The next sections summarize the structure and approaches of CSS.Pome. We will provide some detail regarding the sub-models, which are crucial for understanding the structure, functioning and capabilities of the model. In Appendix A (Table A1), the values of the main parameters are shown, and in the Supplementary Materials (Table S1), further details of the apple and pear trials are reported. Table S2 provides a list of the main symbols and acronyms used in the paper.

The model has three modules (AstroMet, Stand, and Soil). The inputs of AstroMet are weather parameters and data; they generate hourly agrometeorological variables, such as reference evapotranspiration and the potential bare ground and covered soil evaporation values. The module "Stand" inputs stand parameters and data and simulates the ecophysiological processes of the stand, including fruit-size distribution. The "Soil" computes the soil water balance and its components, with the exception of transpiration, which is computed in the module "Stand".

Figure 1 shows a relational diagram with the main modules and procedures that compose the model CSS.Pome.

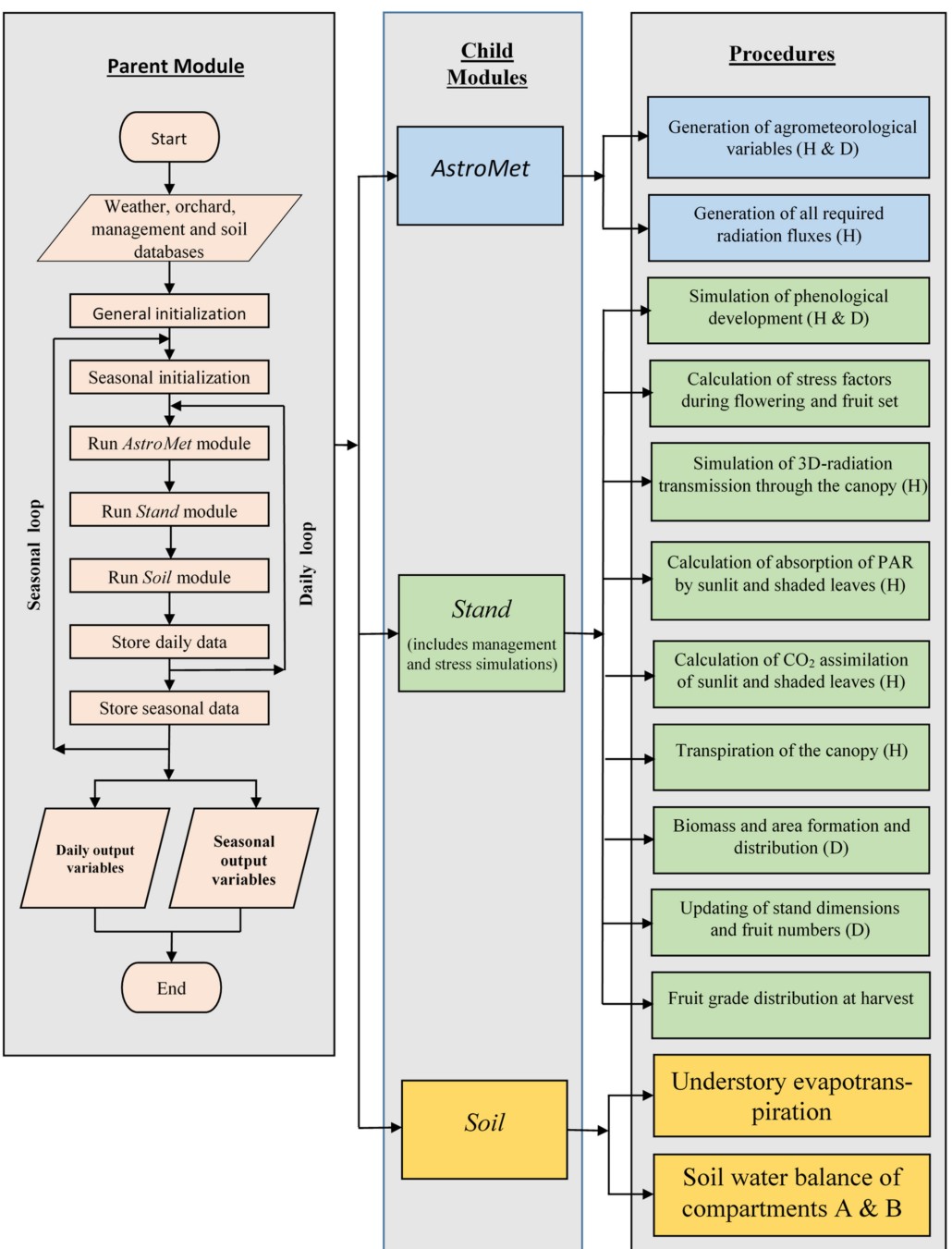

**Figure 1.** Relational diagram of the model CSS.Pome, showing the main modules and procedures and the communication between these structural elements.

2.1.1. AstroMet

This module executes the procedures required to input the forcing variables related to weather, includes the selected number of years of the simulations, and derives all related agrometeorological variables.

The global radiation flux is divided into two wavebands (PAR and NIR) and the corresponding direct (i.e., beam) and diffuse components. Variables related to the orbit of the earth around the sun, such as declination, the equation of time, azimuth, and zenith angle are calculated daily [27,28]. $ET_o$ is calculated using the Penman–Monteith (P-M) approach [17].

### 2.1.2. Stand

Subroutines and functions in this module simulate the development, growth, and production of the orchard stand for a specific number of years. Sub-routines and management inputs are integrated into this module.

Phenological Development

The accurate predictions of bud burst, full flowering, and fruit maturity are essential for model performance.

The flowering prediction uses a sequential model developed by De Melo-Abreu et al. [29,30]. The first phase is the chilling accumulation phase and the second is the forcing phase. The endodormancy is released after enough chilling time has been accumulated and the forcing phase begins. In the forcing phase, there is an accumulation of daily temperature above a base temperature (i.e., thermal time) [10]. Flowering occurs after temperature accumulation reaches an experimentally defined thermal duration. The parameters of this sub-model are determined for each species/variety. In our conditions, the budburst of apple and pear trees occurs about two weeks before full flowering. Hence, the model uses the thermal durations for the forcing phases leading to budburst and full flowering to calculate the dates of occurrence of these phenological events. The prediction of the occurrence of all other development stages uses the saw-tooth model (STM), with corresponding temperature sums, to calculate the dates of occurrence of these stages. STM is similar to the thermal time model, but the air temperature is replaced with an equivalent temperature, to account for the negative effect of supra-optimal temperatures [31].

Radiation Transmission

The model of transmission of radiation through discontinuous, regularly spaced canopies adopts the following basic assumptions and approaches [6]:

(a) The phytoelements of individual tree crowns are contained in volumes bounded by ellipsoidal envelopes. Leaves have a much higher contribution to interception, but the other parts of the plant also intercept radiation. We assume that the shape of the envelope is ellipsoidal, as it can represent a variety of crown shapes that are compatible with single trees or hedgerows.

(b) The ellipsoid is oriented in a right-handed coordinate system. Axes $x$ and $y$ are on the horizontal plane and $z$ is vertical. The positive direction of the $y$-axis points to the lower azimuth angle ($A_z$) of the row of trees, measured counterclockwise from the south. The positive direction of the $x$-axis is perpendicular to the row of plants and 90° away from the positive part of the $y$-axis, measured clockwise. The positive direction of the $z$-axis points to the zenith. The center of the ellipsoid can coincide with the origin of the Cartesian coordinate axes, placed on the surface of the soil, or it can be at a height, $z_c$, above it.

(c) If phytoelements are assumed to be randomly distributed within the volume of the ellipsoidal envelopes, a simple exponential extinction model for beam radiation within the canopy applies [7]. However, most canopies have some form of clustering of leaves, which modifies the radiation interception pattern, resulting from the simplified model. We use the exponential model with an empirical clumping factor, λ = 0.8, that accounts for the effects of leaf-clumping around branches [32–34].

(d) The extinction coefficient of black phytoelements, $K_{eb}(\psi)$, which are usually leaves, uses the ellipsoidal inclination angle distribution [35]. The ellipsoidal model most closely approximates the commonly observed inclination angle distributions of leaves, simply by changing a single parameter (x).

(e) Since the phytoelements in the canopy are not black, we introduce a correction factor, $\sqrt{\alpha}$, where $\alpha$ is the absorptivity of the phytoelements [36]. Therefore, the extinction coefficient of beam radiation in a canopy, considering only "grey" leaves, is $\sqrt{\alpha_L}\, K_{eb}(\psi)$.

The equation that allows simulating the transmission of beam radiation down to a given point is derived from the above basic assumptions and approaches. The fraction of beam radiation transmitted, $\tau_b$, to a specific point within a crown envelope or on the surface of the ground, which is a function of solar zenith angle ($\psi$) and azimuth ($A_z$), is [9]:

$$\tau_b(\psi,\, A_z) = \prod_{i=1}^{n} \tau_{b,\,i}(\psi) = exp\left\{ -\sum_{i=1}^{n} [\sqrt{\alpha_i}\, K_{eb,i}(\psi)\, \mu_i\, \lambda_i\, S(\psi, A_z)\cos\psi] \right\} \quad (1)$$

where $\tau_{b,\,i}(\psi)$ is the fractional transmission of the phytoelements $i$ (i.e., leaves, fruits, twigs, and branches), $\alpha_i$ is the absorptivity of the phytoelements $i$, $K_{eb,i}(\psi)$ is the extinction coefficient of phytoelements $i$, according to the ellipsoidal inclination angle distribution, $\mu_i$ is its area density (m$^2$ of hemi-surface area per m$^3$ canopy volume), $\lambda_i$ is a crown-level clumping factor that accounts for the fact that phytoelements are frequently not randomly distributed inside the envelope, $S(\psi, A_z)$ is the path length through the array of envelopes, which is both a function of zenith angle and azimuth of the sun. In this study, we use the mathematical azimuth definition, which assigns zero degrees to the south direction and considers the angles increasing counter-clockwise from $0°$ to $360°$.

An incident ray can only traverse an envelope or an array of envelopes. The calculation of the path length of a ray through the array of envelopes traversed, $S(\psi, A_z)$, is the sum of the path lengths of each of the envelops, computed in turn. Each envelope can be approximated by a full ellipsoid or an ellipsoid truncated by a horizontal plane located at a height $z_b$ above the ground. CSS.Pome uses the approach first introduced in an earlier study [37] and then extended by other workers [32,38,39]. The mathematical procedure that calculates the path length for a single envelope combines the equation that defines the surface of an ellipsoid with the direction cosines of the radiation ray, terminating at a point on a given horizontal surface. For an array of canopies, the formulation is similar.

The model integrates direct (or beam) and diffuse radiation that reaches a rectangular-elementary mesh on the ground surface, centered in one tree and with $W \times D$ dimensions, where $W$ is the distance between rows and $D$ is the distance between trees. The number of points (nodes) in this elementary mesh is user-defined.

The fractional transmission of direct radiation transmitted to each node is given by Equation (1), while the fractional transmission of diffuse radiation to the nodes, $\tau_d$, is obtained by integrating the transmitted radiation that originates from the whole hemisphere, according to the following equation [9]:

$$\tau_d = \int_0^{2\pi} \int_0^{\pi/2} \tau_b(\psi,\, A_z)\, \sin\psi\, \cos\psi\, d\psi\, dA_z \quad (2)$$

The radiance of overcast skies can be calculated according to the model known as the uniform overcast sky, which considers the radiance of the sky to be uniform and equal to $S_d/\pi$, where $S_d$ is the flux density of diffuse radiation [27].

The remaining equations used to calculate the parameters and variables in Equations (1) and (2) are included in all textbooks on environmental physics (e.g., [27,28]).

The model computes transmissions in the PAR and NIR wavebands, divided into beam unintercepted, for "black leaves", with the beam unintercepted plus scattered, for "grey leaves", and for diffuse radiation. These fluxes are computed for each node and each hour of the day, along with time- and space averages.

The Absorption of Radiation by Shaded and Sunlit Leaves

Daily, hourly data of beam ($I_{ob}$) and diffuse ($I_{od}$) PAR transmission, along with the related variables simulated by the model, are stored in matrices. Now, let us consider the following photon flux densities in the PAR waveband:

- $I_{b,i}$ is the photon irradiance at node $i$ at the surface of the ground, due to the transmission of unintercepted beam radiation (i.e., considering the leaves as "black bodies");

- $I_{bt,i}$ is the photon irradiance at node $i$ at the surface of the ground, due to the transmission of unintercepted beam radiation plus scattered radiation (i.e., considering the leaves as "grey bodies");
- $I_{dt,i}$ is the photon irradiance at node $i$ at surface of the ground, due to the transmission of diffuse radiation, considering these leaves as "grey bodies".

The fractional interceptions related to these fluxes are $\tau_{b,i} = I_{b,i}/I_{ob}$, $\tau_{bt,i} = I_{bt,i}/I_{ob}$ and $\tau_{dt,i} = I_{dt,i}/I_{od}$.

Computation of the average absorption of PAR radiation by shaded (*sh*) and sunlit (*sl*) leaves in discontinuous canopies requires some simplifications. We consider three fluxes that contribute to the total flux density received on the hemi-surface area of leaves in the canopy. The flux density, resulting from the scattering of beam radiation within a continuous canopy, is zero at the top and increases exponentially with the path length through the canopy. Hence, the average flux density can be roughly approximated by an average [27]. Similarly, in the model, the average photon flux density of PAR absorbed on the hemi-surface area of both shaded and sunlit leaves, resulting from scattering, $APAR_{sc}$, is:

$$APAR_{sc} = \frac{0.5 \, \alpha_L \, I_{ob}}{p} \sum_{i=1}^{p} (\tau_{bt,i} - \tau_{b,i}) \tag{3}$$

where $\alpha_L$ is the absorptivity of the leaves and $p$ is the number of nodes with beam interception (i.e., $\tau_{b,i} < 1$). The average PAR that is absorbed by both shaded and sunlit leaves, resulting from the absorption of diffuse radiation, $APAR_d$, is:

$$APAR_d = \frac{\alpha_L I_{od}}{n \, L} \sum_{i=1}^{n} (1 - \tau_{dt,i}) \tag{4}$$

where $n$ is the total number of nodes in the elementary mesh and $L$ is the LAI, averaged over all the terrain.

The average PAR absorbed by sunlit leaves, $APAR_b$, resulting from beam radiation, is:

$$APAR_b = \alpha_L K_{be} I_{ob} \tag{5}$$

where $K_{be}$ is the extinction coefficient, according to the ellipsoidal distribution of the inclination angles of the leaves.

The average PAR absorbed by the hemi-surface area of shaded leaves, $APAR_{sh}$, is:

$$APAR_{sh} = APAR_{sc} + APAR_d \tag{6}$$

and the average PAR absorbed by the hemi-surface area of sunlit leaves, $APAR_{sl}$, is:

$$APAR_{sl} = APAR_b + APAR_{sc} + APAR_d \,. \tag{7}$$

LAI that is sunlit, averaged all over the area of the elementary mesh, $L_{sl}$, is:

$$L_{sl} = \frac{1}{n \, K_{be}} \sum_{i=1}^{n} (1 - \tau_{b,i}) \tag{8}$$

where $n$ is the total number of nodes in the elementary mesh. Shaded LAI, averaged all over the ground, is $L_{sh} = L - L_{sl}$.

Leaf and Canopy Photosynthesis

The photosynthetic $CO_2$ assimilation of leaves uses the biochemical model for $C_3$ species introduced by Farquhar et al. [25], with some modifications [40,41]. The net rate of leaf photosynthesis in µmol m$^2$s$^1$ ($P_n$) is:

$$P_n = min\{W_c, W_j\} - R_d \tag{9}$$

where $W_c$ and $W_j$ are Rubisco-limited and light-limited gross photosynthesis, respectively, and $R_d$ is daytime leaf dark respiration.

The Rubisco-limited gross assimilation rate in µmol m$^2$s$^1$ is:

$$W_c = V_m \frac{p_i - \Gamma_*}{p_i + K_c \left(1 + \frac{O}{K_o}\right)}, \; with \; p_i > \Gamma_* \tag{10}$$

where $V_m$ is the photosynthetic capacity of Rubisco per unit hemi-surface leaf area, $p_i$ is the intercellular $CO_2$ concentration, $K_c$ and $K_o$ are the Michaelis-Menten constants of Rubisco for $CO_2$ and $O_2$, $\Gamma_*$ is the $CO_2$ compensation point in the absence of dark respiration, and O is the oxygen partial pressure. In this paper, $V_m$ is in µmol m$^2$s$^1$ and all other parameters in this equation are in Pa. $V_m$ changes with the species and light environment and is highly correlated with other parameters. Consequently, the values of this parameter for apple and pear crops for shaded and sunlit leaves are inputs to the model [5,42].

Several researchers have reported that the ratio of $CO_2$ concentration in the intercellular spaces and in the atmosphere, $p_i / p_{c,at}$, is fairly constant during the daytime, except for very low levels of conductance when the assimilation rate is quite insensitive to $p_i$. In this model, the ratio $p_i / p_{c, at}$ is 0.7 [43].

The light-limited gross assimilation rate, $W_j$, in µmol m$^{-2}$ s$^{-1}$, is:

$$W_j = J \frac{p_i - \Gamma_*}{4(p_i + 2\Gamma_*)} , \; with \; p_i > \Gamma_* \tag{11}$$

where $J$ is the rate of electron transport per unit of hemi-surface areas of leaves, in µmol m$^{-2}$ s$^{-1}$, which is given by [44]:

$$J = \frac{\alpha_L I \, J_m}{\alpha_L I + 2.1 \, J_m} = \frac{APAR \, J_m}{APAR + 2.1 \, J_m} \tag{12}$$

where $\alpha_L$ is the absorptivity of the leaf, $I$ is the irradiance in µmol m$^{-2}$ s$^{-1}$, $APAR = \alpha_L I$, and:

$$J_m = 29.1 + 1.64 \, V_m$$

following Wullschleger [45].

In Equation (9), $R_d$ is given by [40]: $R_d = 0.015 V_m$ . Temperature dependences of the parameters $V_m$, $\Gamma_*$, $K_c$ and $K_o$ follow the procedure of Bernacchi et al. [46].

Potential-hourly-canopy photosynthesis, $P\prime_P$, in kg $CH_2O$ m$^{-2}$ h$^{-1}$, is:

$$P\prime_P = 1.08 \times 10^{-4}(P_{n,sh} \, L_{sh} + P_{n,sl} \, L_{sl}) \tag{13}$$

where $P_{n,sh}$ and $P_{n,sl}$ are the net assimilation of shaded and sunlit leaves, respectively, in µmol m$^{-2}$ s$^{-1}$, in relation to the unit of the hemi-surface area of leaves. Notice that $P\prime_P$ is termed "potential" since it corresponds to non-limiting water and nutrient conditions.

Potential photosynthesis of the canopy, $P_P$, in kg $CH_2O$ m$^{-2}$ d$^{-1}$, is:

$$P_P = \sum_1^{24} P\prime_P . \tag{14}$$

Maintenance and Growth Respiration

The rate of maintenance respiration depends on the temperature, the plant part under consideration, its composition, and stage. The model calculates the daily maintenance respiration rate of plant part I, $R_{M,i}$, in g $CH_2O$ m$^2$d$^1$, as follows [12,13,47]:

$$R_{M,i} = B_i \times K_{M25, \, i} \, /24 \sum_{j=1}^{24} 2^{(T_j - 25)/10} \tag{15}$$

where $K_{M25,\,i}$ is the daily maintenance respiration coefficient of plant part $i$ at a temperature of 25 °C, in kg $CH_2O$ kg$^1$DW d$^{-1}$, and $B_i$ is the total active dry weight of the plant part. Nevertheless, in this model, the computation of the daily maintenance respiration rate of the leaves, as given by this equation, only includes the night hours in the summation. This is necessary to avoid subtracting twice the glucose lost by leaf respiration during the daytime since net photosynthesis already accounts for the dark respiration of the leaves.

The daily value of maintenance respiration of the stand, $R_M$, is:

$$R_M = \sum_i R_{M,i}. \tag{16}$$

The daily amount of assimilates available for the synthesis of all biomass and the eventual increase of the reserve pool, $\gamma_G$, in kg $CH_2O$ m$^{-2}$ d$^{-1}$, is:

$$\gamma_G = (P_A - R_M + REM) \tag{17}$$

where $P_A$ is the daily-actual-net photosynthesis and $REM$ is the glucose remobilized. Fruit photosynthesis is not considered.

The glucose requirements for the formation of new biomass of the various plant parts are explored in the next section.

Biomass Production and Distribution

CSS.Pome computes the assimilate production and pools on a glucose basis. The allocation of assimilates to the plant parts uses dynamic and static partition coefficients. The DW formation of plant parts results from the amount of glucose allocated to each part and the specific glucose requirement coefficients [14,48,49]. The glucose requirement for the formation of new biomass of the plant part $i$, $G_i$, varies with its composition, but it is fairly independent of temperature.

Fruits are the primary sink of assimilates and their DW increment is calculated first. The potential partition coefficient varies between 0 and 0.7 and is calculated in relation to stage (i.e., thermal time) and sink strength. The increment of daily fruit DW, $\Delta B_{fr}$, is:

$$\Delta B_{fr} = min\left( n_{fr}\, RGR_{P,fr}\, B_{fr}\, \Delta\theta_h\,,\, \frac{0.7\,\gamma_G}{G_{fr}} \right) \tag{18}$$

where $n_{fr}$ is the number of fruits per unit of ground area, $RGR_{P,fr}$ is the potential relative growth rate of an average fruit in relation to hourly thermal time ($\theta_h$), $B_{fr}$ is the fruit DW per unit of ground area, and $G_{fr}$ is the glucose requirement of fruits.

The fraction of assimilates directed to the leaves, branches, trunk, and roots depends on the size of the tree and the developmental stage. The model computes the amount of DW allocated to each compartment, using the differential of the fraction of DW in the compartment in relation to thermal time [5,50]. The daily increment of vegetative DW, $\Delta B_{vg}$, in kg DW m$^{-2}$ d$^{-1}$, is:

$$\Delta B_{vg} = \left( \gamma_G - \Delta B_{fr} G_{fr} \right) / \left( \sum_i \phi_i\, G_i \right) \tag{19}$$

where $\phi_i$ is the fraction of glucose allocated to the vegetative plant part $i$.

The amount of wood removed during pruning and the new dimensions are inputs to the model, but the calculation of the increase in dimensions in the related growth of the different parts of the plant relies on allometric relationships, based on the DW present in these parts [5].

The increment of the hemi-surface area of a plant part results from the product of the specific area of the part and the increment of the part's DW. Specific areas are given as a

function of daily thermal time ($\theta_d$). The specific green-leaf area (*SGA*, in m²kg¹) for apple leaves is:

$$SGA = \begin{cases} A + B\,\theta_d & \theta_d < 1100 \\ 9.95 & else \end{cases} \tag{20}$$

where *A* and *B* are 16.0 and $-0.0055$, respectively. The SGA for pear leaves, in m²kg⁻¹, is:

$$SGA = \begin{cases} A' + B'\,\theta_d + C'\,\theta_d^2 & \theta_d < 1100 \\ 8.12 & else \end{cases} \tag{21}$$

where $A'$, $B'$, and $C'$ are 17.8, $-0.0176$, and $8 \times 10^{-6}$, respectively.

The specific areas of the other plant parts are fixed and have less influence on the extinction of radiation.

Severe Stress Effects

Severe stress affects the number of flowers/fruits, the amount of viable DW, and the green area of the plant parts. The model uses a factor, *FS*, that accounts for temperature and rain effects on flowering and fruit set.

The *FS* is computed during flowering/fruit-set as the product of three factors (*FS1*, *FS2*, and *FS3*). *FS1* accounts for frost damage to flowers/fruits [51]:

$$FS1 = \begin{cases} 0 & T_N \le LT1_{100} \\ \frac{(T_N - LT1_{100})}{(LT1_0 - LT1_{100})} & LT1_{100} \le T_N \le LT1_0 \\ 1 & T_N \ge LT1_0 \end{cases} \tag{22}$$

where $T_N$ is the daily minimum temperature, $LT1_0$ is the critical temperature below which frost damage occurs, and $LT1_{100}$ is the critical temperature that aborts all flowers or damages all fruits.

*FS2* is a daily factor that reduces fruit set, due to a low cross-pollination rate. *FS2* is calculated by:

$$FS2 = 0.7 + 0.3 \times 1/12 \sum_{i=s}^{r} BAI_i$$

where $BAI_i$ is a bee activity index, computed hourly during the daytime ($\approx$12 h). $BAI_i$ takes into account the effects of hourly temperature and rain on bee activity. It is assumed that bees are fully active during the daytime, if air temperature is optimal and there is little or no rain. $BAI_i$ is given by:

$$BAI_i = \begin{cases} \frac{T_h - T_1}{T_2 - T_1} & T_1 \le T_h \le T_2 \\ 1 & T_2 < T_h \le T_3 \\ 1 - \frac{T_h - T_3}{T_4 - T_3} & T_3 < T_h \le T_4 \\ 0 & T_h \langle T_1, T_h \rangle T_4,\ R_h > 0.1,\ W_h > 6 \end{cases} \tag{23}$$

where $T_h$ is the hourly air temperature (°C), $R_h$ is the hourly precipitation (mm), and $W_h$ is the hourly wind speed (m s⁻¹). The values of the parameters in Equation (23), estimated from the literature, correspond to the honeybee, *Apis mellifera* [52,53].

Finally, *FS3* accounts for the effects of heat stress on fruit set. This factor is calculated by:

$$FS3 = \begin{cases} 0 & T_X \ge TH_{100} \\ 1 - \frac{T_X - TH_0}{TH_{100} - TH_0} & TH_{100} \ge T_X \ge TH_0 \\ 1 & T_X < TH_0 \end{cases} \tag{24}$$

where $T_X$ is the daily maximum temperature, $TH_0$ refers to the high temperature below which there is a reduction in the number of fruits, and $TH_{100}$ is the critical temperature above which 100% of the fruits are damaged [54,55].

At the end of the crop season, we use a reduction factor, $F_{ld}$, that affects LAI, thus simulating autumnal leaf drop. This factor is:

$$F_{ld} = \begin{cases} 0 & T_{av} \leq -5 \\ 1 - (5 - T_{av})/10 & -5 < T_{av} < 5 \\ 1 & T_{av} \geq 5 \end{cases}$$

where $T_{av}$ is the average daily temperature (°C).

Potential Transpiration

The potential transpiration of the canopy, $T\prime_P$, on an hourly basis, in mm h$^{-1}$, is:

$$T\prime_P = \frac{0.96 \times 10^6 \left[D + \Delta(T_{can} - T_a)\right] P'_P}{C_a \left(1 - C_i / C_a\right) p_{at}} \tag{25}$$

where $(D = e_s - e_a)$ is the vapor pressure deficit of the air (kPa), $\Delta$ is the slope of the saturation vapor pressure deficit of the air (kPa K$^{-1}$), $T_{can}$ and $T_a$ are the temperature of the canopy and the air, $C_a$ and $C_i$ are $CO_2$ concentrations in the air and in the intercellular spaces ($\mu$mol mol$^{-1}$), $p_{at}$ is atmospheric pressure (kPa), and $P\prime_P$ is the potential hourly-canopy photosynthesis (kg $CH_2O$ m$^{-2}$ h$^{-1}$).

The potential daily transpiration of the canopy, $T_P$, in mm d$^{-1}$, is calculated as:

$$T_P = \sum_1^{24} T\prime_P \tag{26}$$

and actual (i.e., soil water-limited) transpiration of the canopy, on a daily basis, $T_A$, in mm d$^{-1}$, is:

$$T_A = T_P F_w . \tag{27}$$

where $F_w$ is the reduction factor for water stress.

Fruit Growth and Size Distribution

The model outputs both the average fruit fresh weight and dry weight of individual fruits, as well as the fruit size distribution.

Assimilates directed to the fruits along the reproductive crop cycle are determined by thermal time and are limited by the maximum relative growth rate (RGR) of the fruits. The maximum RGR is derived from a logistic equation that is fitted to potential fruit-growth data.

Fruit size is one of the most important parameters to consider when defining the price. The model computes the fruit size distribution at harvest, in relation to the maximum diameter and fruit fresh weight. The fruit size distribution is calculated using the normal distribution, with the population mean ($\mu$) and standard deviation ($\sigma$) as parameters. For both the maximum fruit diameter and fruit fresh weight, the standard deviation is directly proportional to the mean. Hence, $\sigma$ = CV $\mu$, where CV is the coefficient of variance in decimals [56,57].

2.1.3. Soil

This module computes soil water balance and its components, as well as some of the soil properties required for the balance of radiation. Soil depth, density, field capacity, and the permanent wilting point of the soil layers, as defined by the user, are inputs required by this module. The water balance is computed daily; however, some components are initially computed hourly and are integrated later, throughout the day.

Soil Water Balance

In irrigated crops, soil is divided into two multilayer compartments: (i) compartment A extends until the drip line or to a radius that is 40 times the tree-trunk diameter at 20 cm above the ground, whichever is greater, but is limited by the area pertaining to the tree and

receives water from irrigation and precipitation (minus interception); (ii) compartment B, outside the former compartment, receives water from precipitation and, in the case of full-coverage irrigation systems, also from irrigation.

In CSS.Pome, water balance and its components for the two soil compartments are calculated daily and the results are saved separately. The evaporation layer is the top layer, the thickness of which depends on the type of soil. In compartment A, the rooting zone includes this layer and the deeper layers, down to the effective rooting depth, which increases with tree growth. Each compartment of the soil is divided into a user-defined number of homogeneous layers that are characterized by distinctive soil properties. Furthermore, each homogeneous layer is divided into computational sublayers with a user-defined thickness (the default is 0.01 m).

### Runoff, Infiltration, and Deep Percolation

Runoff uses the curve number approach [58]. Soil water distribution and deep percolation into the soil are simulated using a tipping bucket model ("cascading model") [18]. Some versions of this type of model even provide solutions for capillary rise from the groundwater table, without having to resort to differential equations that must be solved numerically. The approach is simple, requires few parameters, and is used by most of the successful models. Moreover, they are often as good as those approaches that are more detailed [18,59,60].

### Soil Evaporation

Soil evaporation follows the approach proposed by the authors of [61]. Potential-wet-bare soil evaporation is calculated directly using the P-M equation with a surface resistance of zero and taking into account the micro-advective effect. When the soil is shaded by the trees, the fractional interception of total radiation, $\tau_t$, multiplies $R_n$ in the energy term of the P-M equation. Potential soil evaporation is reduced by a stress factor that is calculated for the evaporation layer. Soil evaporation is calculated at the same nodes that were set for the computation of radiation fluxes. These nodes are categorized as wet or dry nodes, depending on the type and arrangement of the irrigation devices. Water extraction by orchard floor vegetation is computed using a simplified approach, using a fixed crop coefficient and accounting for one evaporation layer and an understory-rooted layer, with uniform extraction [17]. The net radiation, as in the energy term of the Penman–Monteith formula for the calculation of the reference evapotranspiration of the understory, is computed as described above for the bare soil surface.

### Water Uptake

Water uptake by the trees takes place in Compartment A, where the roots are present. Next, we consider a homogeneous soil but the calculations are similar for multi-layered soils.

Assuming that plants extract more water from sublayers with higher root density and that roots tend to have an exponential distribution, with the depth of soil, the relative extraction potential of a sublayer $I$, at depth $z$ and with a thickness $\Delta z$, is given by the following root distribution function, $F_{r,i}$ [62]:

$$F_{r,i} = \frac{exp(-k_r\,z) - exp[-k_r\,(z + \Delta z)]}{1 - exp(-k_r\,Z_r)} \tag{28}$$

where $k_r$ is the root density coefficient and $Z_r$ is the effective rooting depth. Note that $\sum_i F_{r,i} = 1$, and if all water in the sublayer is depleted, the available water, $Z_r$ must be replaced by $Z_r - \Delta z$, and $F_{r,i}$ must be set to 0.

The root uptake from sublayer $i$, $U_i$, is:

$$U_i = T_P\,F_{r,i}\,F_{w,i} \tag{29}$$

where $F_{w,i}$ is a reduction factor for water uptake, a function of the relative amount of available soil water in sublayer $i$ [15,17].

A reduction factor for water stress, integrating $n$ computational sublayers of the root zone, is:

$$F_w = \sum_{i=1}^{n} F_{w,i} \, F_{r,i} \; . \tag{30}$$

Soil water uptake from the profile, $U$, which is equivalent to actual transpiration, $T_A$, is:

$$U = T_A = F_w \, T_P \, . \tag{31}$$

Evapotranspiration of Orchard Floor Vegetation

It is also possible that orchard floor vegetation can grow. The computation of its contribution to water uptake uses a simplified model with an evaporation layer and one rooted layer, with a uniform distribution of roots in the profile and a user-defined effective rooting depth [17]. In compartment A, the rooted layer extracts water from the profile daily, before the extraction of water by the crop.

### 2.2. Field Experiments

The calibration and validation of the model and its sub-models required extensive experiments, conducted in orchards of the 'Rocha' pear and 'Gala' apple that have been planted in Central and Southern Portugal. Figure S1 contains two maps that show the locations of the experimental orchards. Both regions have a Mediterranean climate, characterized by mild, wet winters and hot, dry summers. According to the Köppen-Geiger climate classification system, the colder and wetter locations in Central Portugal are classified as Csb, while the southern location is classified as Csa [28]. Table 1 shows the average values of some climate variables in a location representative of Central Portugal (Alcobaça) and of the location in Southern Portugal (Torrão).

**Table 1.** The representative climate of the northern and southern experimental sites, Alcobaça (AL) and Torrão (TO). $\overline{T}_N$ is the average of daily minimum temperatures, $\overline{T}_X$ is the average of daily maximum temperatures, $\overline{D}_{dl}$ is the average saturation vapor deficit during the daytime period, and $\overline{S}_t$ is the average of the daily values of global radiation. $R$ and $ET_o$ are the total precipitation and reference evapotranspiration values, respectively. All climate variables were generated by the program "MeuClima&Solo", which is based on recent Climate Normals in Portugal [28].

| Variable | Trimester | | | | | | | | Year | |
| --- | --- | --- | --- | --- | --- | --- | --- | --- | --- | --- |
| | I | | II | | III | | IV | | | |
| | AL | TO | AL | TO | AL | TO | AL | TO | AL | TO |
| $\overline{T}_N$ (C) | 5.9 | 5.6 | 10.5 | 10.6 | 14.0 | 14.6 | 8.2 | 8.0 | 9.7 | 9.7 |
| $\overline{T}_X$ (C) | 15.8 | 16.5 | 20.2 | 24.5 | 25.0 | 31.7 | 18.4 | 19.6 | 19.8 | 23.1 |
| $\overline{D}_{dl}$ (kPa) | 0.65 | 0.74 | 0.84 | 1.52 | 1.22 | 2.61 | 0.80 | 0.97 | 0.88 | 1.46 |
| $\overline{S}_t \left( \text{MJ m}^2\text{d}^{-1} \right)$ | 10.1 | 10.8 | 20.0 | 22.2 | 20.1 | 22.7 | 8.7 | 9.6 | 14.8 | 16.4 |
| $R$ (mm) | 275 | 214 | 164 | 108 | 46 | 34 | 340 | 228 | 825 | 584 |
| $ET_o$ (mm) | 142 | 148 | 309 | 390 | 345 | 462 | 133 | 147 | 929 | 1146 |

Experiment 1 consisted of apple trials, conducted at six locations (A1 to A6) in Central Portugal for four years, starting in 2017. A total of 24 trials were conducted, which included the treatments of tree density and training and pruning techniques. Additionally, for a period of two years, apple phenology was observed in an orchard in Southern Portugal (Torrão).

Experiment 2 consisted of pear trials, conducted in three orchards in Central Portugal (P1, P2, and P3) and cultivated during a five-year period (2006 to 2010). The orchards differed in terms of tree density and the age of the trees, and each orchard included three

treatments of training and pruning. Experiment 3 was conducted from 2012 to 2013 in the same region of Portugal. Orchards P4 to P7 included four treatments that differed in both tree shape and tree density. Orchard P8 was a very high-density orchard and was used to test the effects of tree density. Orchard P9 included four irrigation treatments. Experiment 4 consisted of an orchard planted at a very high density in Southern Portugal in 2015 (P10). This experiment was conducted for a period of six years and was managed according to modern commercial practices [5]. During the growing season, the soils are maintained without soil cover. Table S1 of the Supplementary Materials gives more details of the apple and pear trials.

All crops were managed in accordance with well-established practices [5] and are considered likely to have maintained an appropriate nutritional status. The crops were drip-irrigated along the rows and the percentage of the soil surface that was wetted was between 10% and 15%. The amounts of water applied to the experimental plots were registered and represent the model inputs. Fruit numbers were reduced by chemical thinning in all apple and pear orchards, except in those few years when the flowers or fruits were damaged by frost.

Automated weather stations were available in close proximity to all orchards. All stations measured temperature, humidity, wind speed, global radiation and/or PAR, and precipitation. The soil water content profiles were monitored regularly or continuously in the orchards of Experiments 1 and 2. The instruments used were Diviner 2000 probes (Sentek Pty Ltd., Kent Town, Australia), granular matrix sensors (Watermark$^{®}$), and electrical capacitance probes (Hidrosoph, Évora, Portugal). To study the light environment and estimate the LAI, measurements were taken in most orchards and years, first using a ceptometer placed above the canopy and then on the ground at various distances from the trees and orientations, following a standardized scheme [5]. We used either a Sunscan canopy analysis system (model SS1, Delta-T Devices Ltd., Cambridge, UK) or Sunfleck PAR ceptometers (models SF40 and SF80, Decagon, Pullman, WA, USA). On some apple and pear sites, gas exchange measurements and photosynthesis response curves were measured in the field on clear-sky days, using a LI-6400 (LI-COR, Lincoln, NE, USA) on completely opened leaves [63].

During the crop season, bud burst, full flowering, and industrial maturity dates were recorded in all trials. In some trials, other phenological stages were also observed using the Fleckinger stage scale. Ten years of historical phenological data on the 'Rocha' pear, obtained in the same geographic area in Central Portugal, were also used to calibrate the chilling and thermal time sub-models.

In some trials, the relevant dimensions of the trunk, branches, leaves, and fruits were frequently measured. The specific leaf area (SLA) was also determined in samples distributed throughout the season. In some labeled branches, the fruit load was drastically reduced with the objective of determining the potential fruit growth rate, and the maximum diameter of the few remaining fruits was measured weekly. In general, at the final harvest, all fruits from at least 18 trees per trial were counted and weighed, the maximum fruit diameters were measured, and the firmness and total soluble solids (TSS) measurements were taken on the fruit samples (> 30 fruits per plot).

LAI data were collected from nine trees from an apple orchard (A4 in 2020). All the leaves of each tree were counted; the leaf area of 250 individual leaves per tree was measured using a scanner and the images were processed using ImageJ software.

### 2.3. Data Analysis and Model Performance

Data analysis was performed using Microsoft Excel [64] and R [65]. Various tools, developed by our team, were used to calibrate and validate the sub-models simulating ecophysiological processes, such as chilling requirements, development, leaf assimilation (a batch tool), and the interception of radiation (LAI estimation). These tools are available from the University of Lisbon's Agriculture and Environment Tools webpage [66].

The crop coefficients calculated in this study follow the FAO-56 approach (Allen et al., 1998). The crop coefficient, $K_c$, is $K_c = ET_c/ET_o$, where $ET_c$ is crop evapotranspiration and $ET_o$ is the reference evapotranspiration. The basal crop coefficient, $K_{cb}$, is $K_{cb} = T_P/ET_o$, where $T_P$ is the potential transpiration (i.e., water availability is not limiting transpiration).

The potential fruit size of pear and apple fruits, $W$, expressed as $D_{fr}$ or fruit DW, is described by a logistic curve:

$$W = \frac{\left(W_o\, W_{fr}\right)}{W_o + \left(W_{fr} - W_o\right)\, \exp(-\omega\, \theta_h)} \tag{32}$$

where $W_o$, $W_{fr}$ and $\omega$ are the parameters, and $\theta_h$ is the hourly thermal time.

Relative growth rate, $RGR_{P,fr}$, derived from the previous equation, is:

$$RGR_{P,fr} = \omega\left(1 - \frac{W}{W_{fr}}\right) \tag{33}$$

The calibration process focused on the sub-models of the global model. In general, in the case of pears, this process was carried out using data from the first year of Experiment 2 and Experiment 3. The apple calibration of the sub-models relied, whenever possible, on independent data from unreported orchards cultivated in the same region. Certain processes, like chilling or damage from extreme events, used all the experimental data in both cases. After extensive calibration of such sub-models was undertaken, the model was assembled, coded, and debugged. A slight tuning of the model was required for the pear model, the first that was developed, but not for the apple model.

The validation of the model included the remaining data from the reported experiments. The statistics used to assess model performance were: (1) the statistics of linear regression analysis of predicted versus observed values (intercept, slope, and coefficient of determination); (2) the root-mean-square deviation (RMSD) between the modeled and observed quantities; (3) the modeling efficiency, ME. ME is a statistic that is analogous to the coefficient of determination, which provides an index of performance on a relative scale, where 1 indicates a "perfect" fit, 0 means that the model is no better than a simple average, and negative values indicate a really poor model. In the case of an evidently high performance of the model, some of these statistics are not reported [67–69].

### 3. Results

*3.1. The Validation of Sub-Models*

3.1.1. Phenology

In Central Portugal, full flowering occurred, on average, on DOY 94 and DOY 105 for the 'Rocha' pear and 'Gala' apple, respectively. Budburst of both crops occurred about two weeks before full flowering. The industrial maturity of pears occurred, on average, on DOY 239, while the maturity of apples occurred, on average, on DOY 234.

The model generates thermal time with hourly and daily time steps, and outputs predictions of the dates of bud break, full flowering, and industrial maturity. Table 2 shows the statistics of the validation of the sub-model used to predict the full-flowering occurrence of apple and pear plants [29,30]. For the 'Gala' apple, the number of trials and locations with observed full flowering dates is lower, and observations were fewer for apples than for pears. Hence, the parameters calculated for the process of chilling in 'Rocha' pears are considered to be valid for the 'Gala' apple, with the exception of parameter $a$, which concerns the nullifying effect of high temperatures on the amount of accumulated chilling.

**Table 2.** Parameters of the model for the prediction of data for full flowering (F2) of trees of the 'Gala' apple and 'Rocha' pear [1].

| Crop | N | $T_i$ | $T_o$ | $T_x$ | A | TU | $T_b$ | TT | RMSD | ME |
|------|---|-------|-------|-------|---|----|-------|-----|------|-----|
| **Apple** | 27 | 0 | 6.75 | 24.9 | −1.1 | 750 | 3.5 | 587 | 6.75 | 0.59 |
| **Pear** | 50 | 0 | 6.75 | 24.9 | −0.7 | 1080 | 0 | 709 | 4.86 | 0.69 |

[1] $T_i$ (°C) is the lower cutoff temperature, $T_o$ (°C) is the optimum temperature for chilling accumulation, $T_x$ (C) is the breakpoint temperature above which a constant number of accumulated chilling units (A) are nullified, TU is the number of chilling units (U) required for endodormancy break, $T_b$ (°C) is base temperature for thermal time calculation, and TT (°C d) is thermal duration of the forcing phase [29]. RMSD is the root-mean-square deviation between the modeled and observed DOY of full flowering and ME is the modeling efficiency.

The thermal–time duration from bud burst to full flowering and industrial maturity were evaluated for both apple and pear (see Table A1).

### 3.1.2. Canopy Interception of Radiation and LAI

The global radiation above the canopy was continuously measured during the crop cycle in all trials. In most trials, PAR measurements were performed frequently on the ground in a dense mesh, accompanied by reference measurements over the canopy. Using these data and the corresponding model outputs, we validated the interception sub-model and estimated the LAI.

Table 3 shows the statistics of the validation of the sub-model for the transmission of PAR through the canopies of the experimental apple and pear orchards.

**Table 3.** Statistics of regression analysis of predicted versus observed instantaneous PAR-transmission values in apple and pear orchards [1].

| Crop | N | $\overline{p}$ | $\overline{o}$ | $s_p$ | $s_o$ | a | m | RMSD | ME |
|------|---|------|------|-------|-------|-----|-----|------|-----|
| **Apple** | 1763 | 915.2 | 882.4 | 636.1 | 662.3 | 144.3 | 0.87 | 279.0 | 0.82 |
| **Pear** | 1187 | 1034.5 | 1076.0 | 461.4 | 559.8 | 189.0 | 0.79 | 189.0 | 0.89 |

[1] N is the number of observations, $\overline{p}$ and $\overline{o}$ are the averages of predicted and observed values of transmission, $s_p$ and $s_o$ are the standard deviation of the predicted and observed values, a and m are the intercept and slope of the line, RMSD is the root mean square deviation between the predicted and observed values, and ME is the modeling efficiency. N, m, and ME are unitless and the rest of the quantities are in µmol m$^{-2}$ s$^{-1}$.

The predictions of LAI obtained by the interception sub-model were compared to the direct-destructive measurements of LAI, performed on six separate dates (Sousa, 2013). The RMSD was 0.093 m$^2$ m$^{-2}$ and the ME was 0.99. The measured LAI for the apple orchard (A4 in 2020) was 1.733 ± 0.063 m$^2$m$^{-2}$ and the estimated LAI given by the inverted interception model was 1.720 m$^2$m$^{-2}$.

### 3.1.3. Fruit Growth and Size Distribution

The model calculates the average fruit size and fruit size distribution using both maximum diameter ($D_{fr}$) and fruit weight classes. The function describing the potential relative growth rate ($RGR_{P,fr}$) on a dry weight basis is derived from frequent measurements of the maximum diameter on tagged fruits that are allowed to grow with little or no competition from other fruits in their vicinity. The simulation of fruit size distribution uses the normal distribution, where the parameters are $\mu$ and the standard deviation, $\sigma$ (see Section 2.1.2).

Figure 2 shows the logistic curves fitted to apple and pear fruit growth data. Table 4 contains the parameters of the logistic and fitting statistics of this curve.

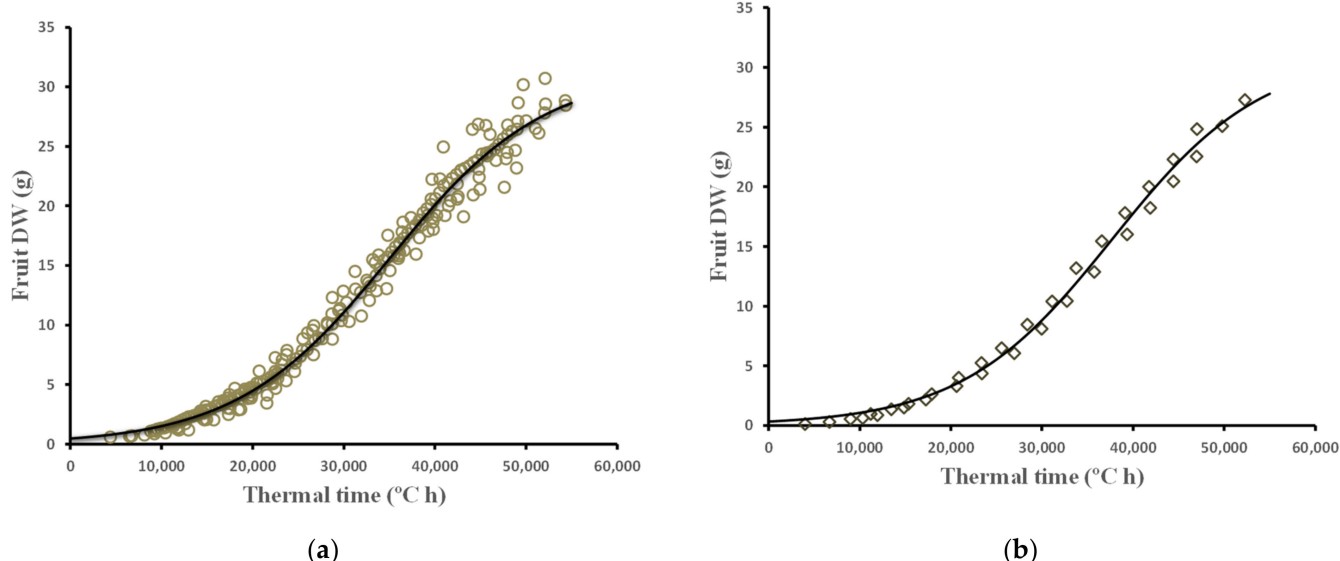

(**a**)                                                                                                      (**b**)

**Figure 2.** Individual fruit DW (g) in relation to the hourly thermal time (°C h) computed after full flowering. Data were collected in apple (**a**) and pear (**b**) experiments. The logistic equations fitted to the data are also shown.

**Table 4.** Parameters of the logistic equation and statistics of the linear regression of predicted on observed values of DW of individual fruits of apple and pear trees grown in the experimental orchards. All fruits monitored were subjected to little or no competition from the fruits in their vicinity [1].

| Fruit | $N$ | $W_o$ (g) | $W_f$ (g) | $\omega$ | $a$ (g) | $m$ | $R^2$ | RMSD (g) | ME |
|-------|-----|-----------|-----------|----------|---------|-----|-------|----------|-----|
| Apple | 230 | 0.49 | 31.39 | $1.18 \times 10^{-4}$ | 0.22 | 0.98 | 0.99 | 1.08 | 0.98 |
| Pear | 34 | 0.32 | 31.22 | $1.21 \times 10^{-4}$ | 0.21 | 0.99 | 0.99 | 0.70 | 1.00 |

[1] $N$ is the number of observations on distinct dates, $W_o$, $W_f$ and $\omega$ are parameters of the logistic equation, $a$ and $m$ are the intercept and slope of the linear regression of predicted on observed values, $R^2$ is the coefficient of determination, RMSD is the root mean square deviation between the modeled and observed fruit DW, and ME is the modeling efficiency.

The fruit-size frequency distribution, defined in terms of fruit diameter and fresh weight, was predicted using the normal distribution equation, considering the CV constant. Half of the data obtained in Experiments 1, 2, and 4 were used for calibration and the remaining data were used for validation. The CV calculated for the maximum-diameter frequency distribution of apples was 0.077 and the CV calculated for fresh weight frequency distribution was 0.246. Similarly, the CVs calculated for pears were 0.074 and 0.224 for the maximum diameter and fresh weight frequency distributions, respectively. Figure 3 shows an example of the agreement between the predicted and observed fruit-size distributions of apple and pear fruits. All fruit samples with an average maximum fruit diameter of between 65 mm and 70 mm were considered for the fruit-size distribution of apples (Figure 3a). Figure 3b includes fruits with an average maximum fruit diameter of between 60 mm and 65 mm. Table 5 shows the results of the validation of the sub-model for the size distribution of all apples and pears. The limits of the classes shown related only to the maximum fruit diameter, since the limits in fruit fresh weight resulted in almost identical values of the average number of fruits, due to the close relationship between the two variables.

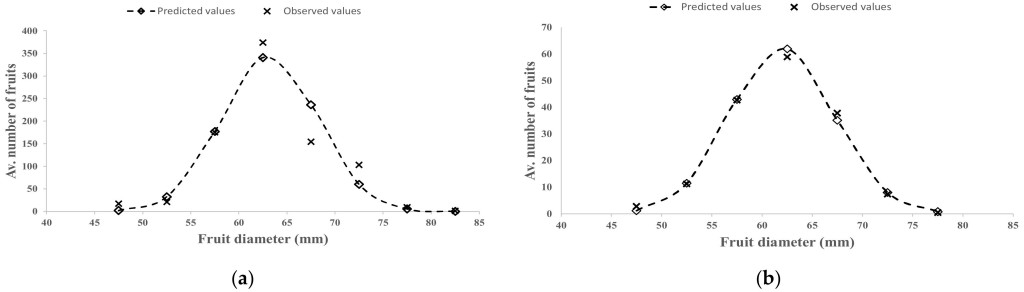

(**a**)         (**b**)

**Figure 3.** Example of the agreement between the predicted and observed fruit-size distribution, expressed as fruit diameter, from apple (**a**) and pear (**b**) experiments. The average fruit diameters of the samples, as used in this example, were between 65 and 70 mm for apple samples and between 60 and 65 mm for pear samples.

**Table 5.** Statistics of the predicted versus observed number of fruits in fruit-size classes. Class boundaries are the maximum fruit diameters in millimeters [1].

| Crop | $N$ | $\bar{p}$ | $\bar{o}$ | $s_p$ | $s_o$ | $a$ | $m$ | $R^2$ | RMSD | ME |
|------|-----|-----------|-----------|-------|-------|-----|-----|-------|------|-----|
| Apple | 84 | 144.2 | 142.5 | 138.1 | 149.7 | 21.0 | 0.86 | 0.86 | 21.1 | 0.88 |
| Pear | 900 | 27.0 | 27.0 | 37.1 | 37.2 | 0.47 | 0.98 | 0.97 | 6.2 | 0.97 |

[1] $N$ is the number of values in the regression of predicted on observed numbers of fruits in the classes, $\bar{p}$ and $\bar{o}$ are the averages of predicted and observed fruit numbers, $s_p$ and $s_o$ are the standard deviations of the predicted and observed fruit numbers, $a$ and $m$ are the intercept and slope of the regression predicted for observed fruit numbers, $R^2$ is the coefficient of determination, RMSD is the root mean square deviation between the predicted and observed fruit numbers, and ME is the modeling efficiency.

### 3.2. Validation of the Global Model

The model accounts for potential and water-limited conditions, cold and heat stress during flowering, and frost damage to flowers and small fruits. Simulations refer to apple and pear orchards with several management treatments, such as training and pruning, and tree density. We assumed that the input variables to the model were likely to grasp the essence of these treatments, thus allowing the pooling of all orchards of a given species. The main objectives of the model are to predict the pome fruit productivity of orchards and the fruit size distribution. Figure 4 shows simulated versus observed productivity values for all apple and pear trials and the 1:1 line. Notice that two crops of pear had almost no yield, due to prolonged low temperatures and rain during flowering. Table 6 shows the performance statistics of the global model for apple and pear orchards. The parameters of the regression of predicted data on observed productivities show a similar tendency, which consists of noticeable positive intercepts and slopes that are lower than unity. However, the RMSD is lower than 13.39 t ha$^{-1}$, while ME is higher than 0.85.

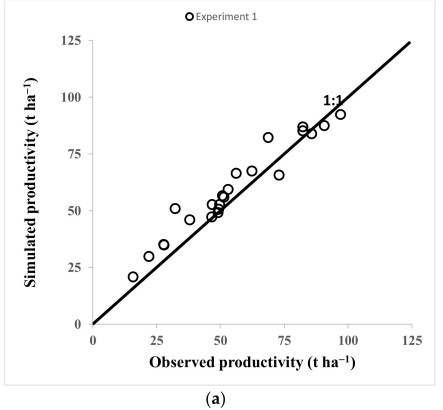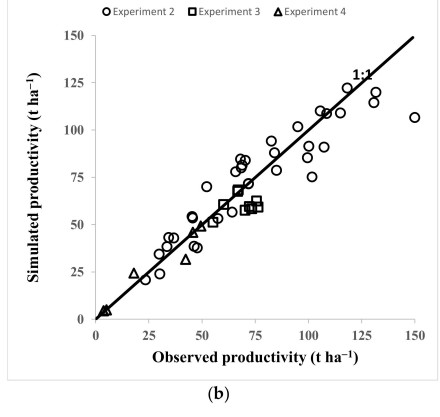

(**a**)         (**b**)

**Figure 4.** The predicted versus observed productivities of apple (**a**) and pear (**b**) crops, and their relation to the 1:1 line.

**Table 6.** Statistics of the predicted versus observed productivities of apple and pear crops.[1].

| Crop | $N$ | $\overline{p}$ | $\overline{o}$ | $s_p$ | $s_o$ | $a$ | $m$ | $R^2$ | RMSD | ME |
|------|-----|-----|-----|-----|-----|-----|-----|-----|------|-----|
| Apple | 23 | 59.2 | 54.7 | 19.59 | 22.09 | 12.05 | 0.86 | 0.94 | 7.17 | 0.90 |
| Pear | 51 | 67.2 | 70.3 | 29.15 | 33.99 | 11.42 | 0.79 | 0.83 | 13.39 | 0.85 |

[1] $N$ is the number of values in the regression of predicted and observed productivities, $\overline{p}$ and $\overline{o}$ are the averages of the predicted and observed values, $s_p$ and $s_o$ are the standard deviations of the predicted and observed values, $a$ and $m$ are the intercept and slope of the regression of predicted on observed values, $R^2$ is the coefficient of determination, RMSD is the root mean square deviation between the predicted and observed values, and ME is modeling efficiency. $N$, $m$, $R^2$, and ME are unitless, while the rest of the quantities are in t ha$^{-1}$.

## 4. Discussion

The accurate prediction of phenological events, especially the timing of budburst, flowering, and maturity, is essential for high model performance. The sequential model for predicting the occurrence of full flowering yields differences between the modeled and observed values of this event, which are less than a week on average, both for apple and pear crops, since the RMSD is 6.8 and 4.9 days for apple and pear, respectively (Table 2).

The interception sub-model yielded ME values of 0.82 for PAR transmission through apple canopies and 0.89 for pear canopies (Table 3). These and the rest of the validation results show that the performance of this sub-model is high in both cases. The mean values of the modeled and observed radiation fluxes, transmitted through the canopy, and the variability associated with these data are similar. The values of $a$ and $m$ indicate that there is a small overestimation for low values of transmitted PAR and a small underestimation for higher values. Apparently, there is a compensation effect, corroborated by the similar values of $\overline{p}$ and $\overline{o}$, leading to the accurate predictions of LAI that were previously reported (see Section 3).

The logistic equation, Equation (32), describes the potential growth of individual fruits in relation to thermal time very effectively. The model computes the daily values of RGR for Equation (33), which provide an upper boundary to the potential daily growth of fruits. In cases where the assimilate available for translocation to the fruits is less than the daily potential growth, the actual growth is the amount of assimilate available.

In a favorable physical and biological environment, well-managed trees growing without water and nutrient limitations are likely to have high yields. Potential fruit growth can occur in a fraction of the total number of fruits, but the other fruits will grow more slowly. Apparently, the fraction that offers potential growth increases as the crop load decreases, and vice versa. Our results corroborate previous reports showing that fruit size distribution approaches normality [56,70]. This allows a direct distribution of the fruits according to user-defined fruit-size classes. The statistics presented for apples and pears indicate that this approach is suitable for predicting the distribution of fruit sizes by grade classes once the average fruit size is known.

The global model accurately predicted the fruit productivity indicated by the ME of 0.90 and 0.85 for apple and pear crops, respectively (Table 6).

The values of key output variables of the model were confronted with the results obtained in studies performed by other researchers, but the internal coherence of the model was not refuted. Further examples follow.

In general, $C_3$ plants have a transpiration ratio of around 500 (i.e., 500 molecules of water are lost for every molecule of $CO_2$ fixed by photosynthesis). When WUE is expressed as the inverse of the transpiration ratio, the WUE corresponds to 3.33 g of fixed glucose per kg of water transpired. However, the value of WUE is not conservative and will vary with certain plant and environmental factors. For example, it decreases as the VPD or heat stress increases and it increases as $CO_2$ concentration in the air or stomatal conductance increases. The seasonal values of the WUE of $C_3$ plants, under Mediterranean conditions, are often in the range of 3 to 6 g of fixed glucose per kg of water transpired. Lower values are usually associated with spring crops and winter crops growing into late summer. Winter cereals that stop growing at the beginning of summer have higher values of WUE [71,72].

In this study, we computed the seasonal values of WUE$_{ab}$ as the ratio of DW produced above-ground and water used in transpiration for comparison with values obtained by other researchers under similar conditions. Our values of WUE$_{ab}$, corresponding to mild or no water stress conditions, are $5.7 \pm 0.2$ and $5.8 \pm 0.2$ g DW per kg of water transpired. These values are compatible with the corresponding values found in the literature for apples [73] and pears [5,74].

In the apple orchards, the simulated values of crop coefficient ($K_c$) and basal crop coefficient ($K_{cb}$) in the mid-season were $0.85 \pm 0.3$ and $0.57 \pm 0.3$, respectively. The calculation of these coefficients was restricted to unstressed orchards, with trees of three years old and older. Marsal et al. [75] reported values of $K_c$ for mid-season, ranging from 0.55 to 0.96, corresponding to the first eight years of growth of the trees. The range of our values of $K_c$, considering trees aged one to eight years old, after planting was between 0.67 and 0.93. The average mid-season value of $K_c$ (0.85), as presented above, is similar to that reported by Marsal et al. [75] for apple trees of about the same age and size, grown in Spain under climate conditions and with management analogous to our trials. Apple orchards in South Africa, kept under Mediterranean conditions, had an average mid-season $K_c$ of 0.79 [76]. This report shows that apple orchards in Israel, Spain, and Australia yielded similar values of mid-season $K_c$. The average values of mid-season $K_c$ were 0.60 and 0.77 in two consecutive years, measured two and three years after planting. The experiment was conducted in an apple orchard in Chile under a Mediterranean climate; the water application was by drip irrigation [77]. Orchards grown in sub-humid climates, moreover, those using irrigation systems that wetted all the ground, have higher mid-season $K_c$ values due to the higher soil evaporation and higher stomatal conductances [78]. Mid-season $K_{cb}$s averaged 0.61 in an experiment conducted in Chile in an apple orchard [77].

In the pear orchards, the simulated values of crop coefficient ($K_c$) and basal crop coefficient ($K_{cb}$) in the mid-season were $0.81 \pm 0.3$ and $0.67 \pm 0.2$, respectively. The calculation of these coefficients was restricted to unstressed orchards, with trees of three years old and older. Most mid-season $K_{cb}$s reported by other workers for pear orchards in a comparable climate, stage, and management are similar to the values computed in this study. Girona et al. [79] obtained a $K_c$ for the mid-season of 0.85 for three-year-old pear trees grown in a large lysimeter in the middle of an orchard. Goodwin et al. [80] calculated the $K_{cb}$ from a pear orchard with a fraction of intercepted PAR that was equal to 0.61. The mean value of orchard $K_{cb}$ was $0.55 \pm 0.012$. Conceição et al. [81] measured a lower mid-season $K_c$ value, using the eddy-covariance method, in an experiment conducted during one season in a 5-year-old orchard in Central Portugal. The $K_c$ values obtained in mid-season averaged 0.53, ranging from 0.46 to 0.63. Precipitation during the relevant period was negligible and the soil had no ground cover under the canopy. The value of ground cover shown by the crop was low (35%), compared to the ground cover of most of the orchards used in our $K_c$ and $K_{cb}$ calculations (> 50%).

One of the main objectives of this study was to provide a system that assists growers to decide the optimal crop load of a given apple or pear orchard, in order to obtain the optimal combination of productivity and fruit-size grade. We have always been of the opinion that such a system can only be valid for any site and orchard if it is supported by a simulation model, provided that the required parameters are input. Next, we present an example based on a hypothetical apple orchard, with the characteristics, management, and environment of the "Acipreste" site. The predictions corresponded to the year of the sixth harvest of the orchard, and irrigation was applied at a potential level.

Ideally, weather inputs should consist of a representative series of weather data for the same location, excluding the years of rare events (or specifically including them). However, this example uses only four years of weather data, collected at the location. Figure 5 shows the average predicted productivity and fruit size in relation to crop load for the specific orchard in the sixth year of harvest. The normal distribution, with the parameters presented above, predicts the fruit-size distribution. We are working on a tool designed to automate the entire process and provide the results in both graphical and tabular forms.

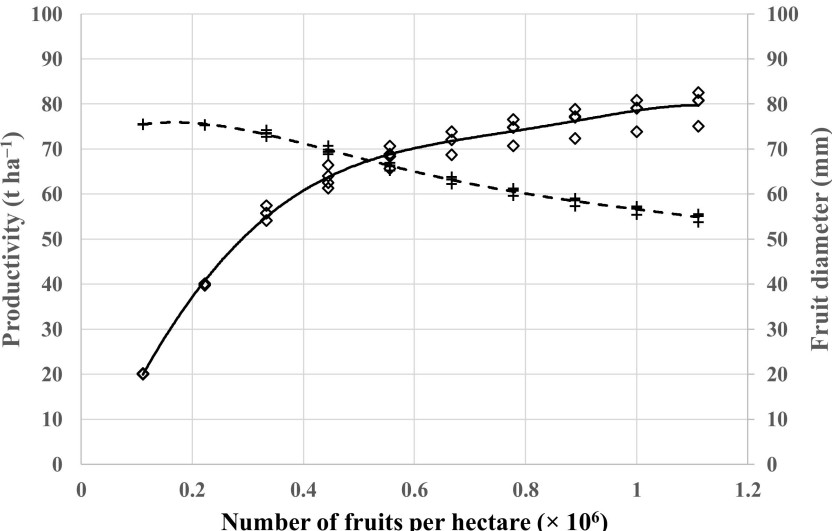

**Figure 5.** Example of simulated productivity (◊) and fruit diameter (+) in relation to crop load in an apple orchard. The site is in Acipreste, the trees are five years old at the beginning of the hydrological year, and the parameters are actual parameters measured in the orchard. The lines show polynomial fits to the simulated points.

## 5. Conclusions

CSS.Pome met all the intended specifications. The model predicted the productivity of apple and pear crops very well (ME ≥ 0.85) and the fruit-size distribution was close to the observed values (ME ≥ 0.88). The model requires a small number of parameters that are easy to determine, the program is user-friendly, and all the tests performed showed that the internal consistency of the model was high.

The model provides a system to help growers to determine the optimal crop load for a given apple or pear orchard. Farm advisers and advanced growers can use the model to create tables or charts for specific orchards that allow the determination of the ideal crop load to achieve maximum productivity, with fruits of the required size grade.

We identified some areas to further improve CSS.Pome. Experiments could be conducted, for example, to support the implementation of a nitrogen balance sub-model to simulate the effects of nitrogen shortage on productivity, quality, and crop load. More detail could be introduced regarding high-temperature and high-radiation stresses, in order to avoid or minimize sunscald. Finally, these databases of apple and pear parameters could be expanded to include other cultivars.

**Supplementary Materials:** The following supporting information can be downloaded at: https://www.mdpi.com/article/10.3390/horticulturae8100873/s1, Table S1: Details of experimental apple and pear orchards; Table S2: List of main symbols and acronyms in this paper; Figure S1: Location maps of the experimental orchards.

**Author Contributions:** Conceptualization, M.L.S. and J.P.D.M.-A.; methodology, M.L.S., J.P.D.M.-A. and A.R.; software, J.P.D.M.-A.; validation, M.L.S. and J.P.D.M.-A.; formal analysis, M.L.S., J.P.D.M.-A., M.G. and A.R.; investigation, M.L.S., J.P.D.M.-A., D.F., M.G., J.P.L., and A.R.; resources, M.L.S., J.P.D.M.-A. and C.M.O.; data curation, M.L.S., J.P.D.M.-A., M.G., J.P.L., and A.R.; writing—original draft preparation, M.L.S. and J.P.D.M.-A.; writing—review and editing, J.P.D.M.-A. and C.M.O.; supervision, J.P.D.M.-A.; project administration, M.L.S., J.P.D.M.-A. and C.M.O.; funding acquisition, M.L.S. and J.P.D.M.-A. All authors have read and agreed to the published version of the manuscript.

**Funding:** This work is supported by the project OPTIMAL (Optimização, Maçã, Alcobaça)-PDR2020-1.0.1-FEADER-031444, promoted by PDR2020 and co-financed by EAFRD, within the scope of Portugal2020.

**Data Availability Statement:** Not applicable.

**Acknowledgments:** Many colleagues who are not included in the authorship of this paper and the technicians contributed in the last two decades for the data set used to develop sub-models. We thank them for their work, advice and support. COTHN, through Carmo Martins, was responsible for numerous events and other outreach efforts that benefited our research.

**Conflicts of Interest:** The authors declare no conflict of interest.

## Appendix A

**Table A1.** List of the main parameters of CSS.Pome and their values. EE-BB = end of endodormancy to budbreak; BB-FF = budbreak to full flowering; FF-MT = full flowering to fruit maturity; Lf = leaf; Byr = current-year stem; Br = branch (diameter $\leq$ 10 mm); Tr = trunk (diameter > 10 mm); Rt = root; Fr = fruit.

| Parameter | Symbol | Unit | Value | Source |
|---|---|---|---|---|
| Absorptivity of NIR (Lf) | $\alpha_{L,NIR}$ | dimensionless | 0.15 | [27] |
| Absorptivity of PAR (Lf) (Equation (1)) | $\alpha_L$ | dimensionless | 0.85 | [27] |
| Atmospheric concentration of $CO_2$ | $C_a$ | μmol mol$^{-1}$ | 413 | [24] |
| Attenuation coefficient of wind in the canopy | $a$ | dimensionless | | [27] |
| Base temperature for EE-BB, BB-FF (Apple) | $T_b$ | °C | 3.5 | * |
| Base temperature for EE-BB, BB-FF (Pear) | $T_b$ | °C | 0 | [6] |
| Base temperature for FF-MT | $T_b$ | °C | 0.4 | [6] |
| Chilling units for dormancy break (apple) | $TU$ | U | 750 | * |
| Chilling units for dormancy break (pear) | $TU$ | U | 1149 | [6] |
| Chilling units nullified above Tx (apple) | $A$ | U | -1.1 | * |
| Chilling units nullified above Tx (pear) | $A$ | U | -0.7 | [6] |
| Clumping factor (Lf) (Equation (1)) | $\lambda$ | dimensionless | 0.8 | [32–34] |
| $CO_2$ compensation point in the absence of dark respiration (25 °C) | $\Gamma_*$ | Pa | 4.28 | [46] |
| Depth of surface soil evaporation layer | $Z_e$ | m | 0.10 | [6] |
| Ellipsoidal-leaf-angle distribution parameter (apple) | $x$ | dimensionless | 3.5 | [34] |
| Ellipsoidal-leaf-angle distribution parameter (pear) | $x$ | dimensionless | 3.5 | [34] |
| Evapotranspiration depletion factor | $p$ | dimensionless | 0.5 | [17] |
| Fraction of reserves in stems and roots at budburst | $F_{0,res}$ | dimensionless | 0.1 | [74] |
| Global radiation to PAR (conversion factor) | $F_{SP}$ | μmol J$^{-1}$ | 4.55 | [28] |
| Glucose requirement of apple Lf, Byr, Br,Tr, Rt, Fr | $G_i$ | kg $CH_2O$ kg$^{-1}$ DW | 1.43, 1.34, 1.39,1.35, 1.34, 1.13 | *, [48,49] |
| Glucose requirement of pear Lf, Byr, Br,Tr, Rt, Fr | $G_i$ | kg $CH_2O$ kg$^{-1}$ DW | 1.43, 1.34, 1.39,1.35, 1.34, 1.12 | *, [48,49] |
| Initial fraction of coarse-root DW | $cRti$ | dimensionless | 0.16 | *, **, [49] |
| Initial fraction of fine-root DW | $fRti$ | dimensionless | 0.11 | *,**, [49] |
| Initial fraction of stems DW | $Sti$ | dimensionless | 0.73 | *, **, [49] |
| Lower cutoff temperature for chilling accumulation | $T_i$ | °C | 0 | [29] |
| Maintenance respiration coefficient (Lf, Byr, Br,Tr, Rt, Fr) | $K_{M25,\ i}$ | g $CH_2O$ kg$^{-1}$ DW d$^{-1}$ | 13, 9,3,0.3,3,2 | [3] |
| Michaelis-Menten constant of Rubisco for $CO_2$ (25 °C) | $K_c$ | Pa | 40.49 | [46] |
| Michaelis-Menten constant of Rubisco for $O_2$ (25 °C) | $K_o$ | Pa | $27.84 \times 10^3$ | [46] |
| Optimum temperature for chilling | $T_o$ | °C | 6.75 | [29] |
| Optimum temperature for EEBB, BB-FF, FFMT | $T_o$ | °C | 25 | [6] |

**Table A1.** *Cont.*

| Parameter | Symbol | Unit | Value | Source |
|---|---|---|---|---|
| Parameters Equation (22) for apple | $LT1_0$, $LT1_{100}$ | °C | -2.7, -4.9 | [51] |
| Parameters Equation (22) for pear | $LT1_0$, $LT1_{100}$ | °C | -2.7, -7.4 | [52,53] |
| Parameters Equation (23) for apple and pear | $T_1$, $T_2$, $T_3$, $T_4$ | °C | 12, 16, 30, 45 | [52,53] |
| Parameters Equation (24) for apple and pear | $TH_0$, $TH_{100}$ | °C | 30, 45 | *, [54,55] |
| Photosynthetic Rubisco capacity of apple leaf (25 °C) (sunlit Lf, shaded Lf) | $V_m$ | µmol m$^{-2}$ s$^{-1}$ | 175, 142 | [42] |
| Photosynthetic Rubisco capacity of pear leaf (25 °C) (sunlit Lf, shaded Lf) | $V_m$ | µmol m$^{-2}$ s$^{-1}$ | 202, 174 | ** |
| $Q_{10}$ for maintenance respiration | $Q_{10}$ ($R_M$) | dimensionless | 2 | [47] |
| Rainfall interception coefficient | $I_r$ | mm/LAI | 0.15 | [6] |
| Reflection coefficient of dry soil (PAR, NIR, LW) | $s,d$ | dimensionless | 0.15, 0.30, 0.04 | [6] |
| Reflection coefficient of wet soil (PAR, NIR, LW) | $s,w$ | dimensionless | 0.08, 0.16, 0.04 | [6] |
| Runoff parameter | $S$ | dimensionless | 0.14 | [58] |
| Thermal duration for EE-BB, BB-FF, FF¬MT (Pear) | $TD$ | °C d | 466, 175, 1763 | **, [6] |
| Thermal duration for EE-BB, BB-FF, FF-MT (Apple) | $TD$ | °C d | 449, 138, 1697 | * |
| Upper breakpoint temperature for chilling | $T_x$ | °C | 24.9 | [29] |
| Upper cutoff temperature for EE-BB, BB-FF, FF-MT | $T_x$ | °C | 35 | [6] |

\* Data from Experiment 1 and current analysis. ** Data from Experiments 2–4 and current analysis.

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
