# Peer review of "Apple and Pear Model for Optimal Production and Fruit Grade in a Changing Environment"

_horticulturae, doi:10.3390/horticulturae8100873_

Round 1

Reviewer 1 Report

This study simulated the production and fruit grade of apples and pears through phenological models, which is a topic worth studying. The study was based on a large number of experiments and had a long observation time, and the crop yield and fruit diameter were predicted by the models with high accuracy. In addition, it had some practical application values,such as using the model to predict fruit size distribution by grade category and help growers determine the ideal crop load, et al.. Besides, the model can be used for predicting crop phenology and guiding agricultural production. However, I have the following major concerns need to be addressed: 

(1) Is the model used in this manuscript an improved and optimized model or just a simple application of an existing model? 

(2) There were differences in the simulation results of the study on unconsctioned orchards and orchards with lower tree age, and the follow-up study need optimize the crop model; (3) The manuscript can be refined, and the content is too complicated. 

I have also some minor points:

1. Why do the parameters in the second "Stand" module also contain radiation variables?

2. Line 66, add an reference for the introduction for this model.

3. There are too many contents for each parameter in the "Methods". It is recommended to make a dramatic cut for section 2.1.

4. This model contains so many parameters, how to simulate? How to quantify the relationship between parameters? How to combine? At least you should give sufficient introductions to these problems.

5. Are all experiments with apples or pears the same variety? This needs to be emphasized in the 2.2 section of the manuscript.

6. I strongly recommend the authors add a map to the manuscript, and mark the basic information of the weather stations, the experimental stations and the orchards on the map. 

7. In section 3.1.3, the formula should not appear in the "result" section. It can be described in the second part.

8. In Figure 1, the ordinate value is wrong, please modify it.

Reviewer 2 Report

This manuscript presents a model for the simulation of productivity and fruit grade of apple and pear orchards under potential and water limited conditions based on a large amount of measurements in Portugal. A model description was given and the submodels were calibrated and evaluated. This manuscript is well written and only needs some minor grammatical corrections. Since this model is rather complex and complete a conceptual scheme, flow diagram should be added showing how the different submodels connect. This would enhance our understanding of the model. I insist on adding the supplementary materials in the main manuscript as appendices. To my opinion these two tables are far too important for understanding the model. Since also a lot of parameters and abbreviations are used, an additional list in the appendices would make it easier for readers. This list should complement the parameters of Table S2 (no need to repeat parameters listed in this table).

The “choice of training system” is not well explained and I do not know what the authors are trying to explain. Please elaborate;

L16, 33: It is annoying to read the same sentence twice in the first line of abstract and Introduction. Slight rephrasing is required.

L59: “to resort” is not an appropriate word; Perhaps better is “to have the support of”;

L88: should be “The inputs of AstroMet are …”;

Eq. (2): What is “sen” in this equation? Should it not be “sin” (sine);

L277: from this sentence I derive that DW means “dry weight”, but I believe that this abbreviation was not mentioned/explained before;

Eq (27): What is “Fw”? I believe that this parameter was not mentioned earlier in the text. Water stress factor?

L377-378: should be “… the normal distribution with the population mean and standard deviation as parameters.”;

L469: should be “at six locations”;

L487: rephrase “the irrigations wet about 10-15% of the soil.” This is not a proper sentence;

L517: Should be “LAI data were collected in…”;

L526: “follow THE FAO-56”;

L536: what do you mean with the “global model”?

L550: DOY 94 and 105 are derived from field observations? If so, please add in text;

Table 4 should be placed under the Table captions and above the legend;

Figure 2, Figure 3. reformat to remove the lines that make a frame; It make a figure too busy;

Table 5: add the R2 value of the regression lines;

Table 6: add the two R2 values of the regression lines;

The section with the conclusions should be extended to future research; What should be done in the future to improve the apple and pear model? Give perspectives.

Table S2: Make the column with units a bit bigger so the units on glucose requirements are on one line;

For some parameters no units are given: add [-] if appropriate;

Round 2

Reviewer 2 Report

Thanks for the improvements. With the flowchart the model design is much more clear. 

Please review the tables in the appendix. The format is mess (copying went not ideally, I think).  

Author Response

Dear Reviewer,

There is a software problem with Table A1. I also had this problem once. I suspect it is either a problem of the process to view the “Track changes” markings or the fact that the table may have been too wide.

I send now a version with all changes accepted, including Table A1, and made a minor reduction on the width of the table. If possible, I will upload also a PDF version. I will try to append the PDF in the place where I should send the reply to your comments

Without trying to be a referee of a referee, I would like to thank you very much for your great work that improved a lot our manuscript.

Best regards,

JPMA
